# Dental Status and Oral Health Behaviors of Selected 45–74-Year-Old Men from Northeastern Poland

**DOI:** 10.3390/ijerph20116005

**Published:** 2023-05-30

**Authors:** Malgorzata Pawinska, Anna Kondrat, Jacek Jamiolkowski, Elzbieta Paszynska

**Affiliations:** 1Department of Integrated Dentistry, Medical University of Bialystok, M. Sklodowska-Curie Str. 24A, 15-276 Bialystok, Poland; 2Dental Clinic—NZOZ Dent-Plast, Radzyminska Str. 5, 15-863 Bialystok, Poland; 3Department of Population Medicine and Lifestyle Diseases Prevention, Medical University of Bialystok, Waszyngton Str. 13A, 15-269 Bialystok, Poland; 4Department of Integrated Dentistry, Poznan University of Medical Sciences, Bukowska Str. 70, 60-812 Poznan, Poland

**Keywords:** dental status, DMFT, missing teeth, adult, oral health behavior, smoking, oral hygiene

## Abstract

The aim of this study was to assess the dental status and oral health behaviors of selected 45–74-year-old men from northeastern Poland. A total of 419 men were included. A questionnaire on demographic data, socioeconomic status and oral health behaviors was conducted. Dental caries experience (DMFT index), oral hygiene (AP index) and a number of edentulous subjects were evaluated clinically. More than half of the respondents (53.2%) brushed their teeth once a day. Nearly half of respondents (45.6%) reported for check-up visits more rarely than once in two years. Active nicotinism affected 26.7% of males. The prevalence of decay, the mean DMFT, the mean API and the prevalence of edentulism were, respectively, 100%, 21.4 ± 5.5, 77% and 10.3%. Increased DMFT values and MT were significantly correlated with older age (*p* < 0.001). Subjects of high educational status showed significantly lower values of DMFT and MT (*p* < 0.001). An increase in per capita family income was accompanied by a significant decrease in the API (*p* = 0.024), and an increase in DMFT (*p* = 0.031). This study demonstrated low health awareness and unsatisfactory dental status among the examined males. Dental and oral hygiene status were associated with sociodemographic and behavioral determinants. The poor oral health condition of the study population indicates the need to intensify pro-health education among seniors about oral care.

## 1. Introduction

Dentition status is an essential element of oral health. If dentition is in a poor condition, it can unfavorably affect patient quality of life, especially in terms of physical pain, functional limitations, and social disability [1,2]. In addition, a study indicates that the poor dental condition (tooth loss) may be an early indicator of accelerated aging [3].

According to a recent WHO report, caries remain the most widespread oral condition affecting people of all ages. Caries continue to be a global public health challenge [4]. Poland is one of few European countries where the prevalence of caries is still very high—almost 100% in the adult population—and epidemiological data have never been satisfactory [5]. Chronic caries in adults may lead to substantial hard tissue loss and attenuate tooth longevity and treatment prognosis. In the elderly, the most significant type of dental caries is root caries, associated with exposure of the root cement by gingival recession, heavy consumption of fermentable carbohydrates, and the probability of xerostomia caused by age-related salivary alterations and multimedication [6]. A complication of caries is pulp inflammation, which can spread infection to surrounding tissues and lead to serious systemic consequences. Along with dental caries, there are several other oral diseases that are common in older adults—periodontal disease, dry mouth, lip and oral cancer, and extensive tooth loss. Untreated tooth decay and periodontal disease are not only the cause of premature tooth loss, but even lead to toothlessness. This contributes to a decrease in elderly patients’ well-being and self-esteem, and worsens their social functioning [7]. Moreover, oral infections (especially periodontitis) may be considered an independent risk factor for some diseases including diabetes, cardiovascular disease, pulmonary infections, kidney disease, and dementia [8,9].

Recently, the world’s elderly population has increased significantly and prognoses indicate that this trend will continue over the next decades [10]. Analysis of demographic prognoses for 2014–2050 shows that in approximately 2050, every second urban-dwelling man will be over 51, while every second woman will be over 56 in Poland. The rural population will be a few years younger. In addition, life expectancy will increase over the prognosis period, reaching values higher in 2050 than those recorded in 2013 by nine years for men and six years for women. Men will live to 82 years old (73 in 2013), and women to 87 years old (81 in 2013) [11].

Sex can affect oral health due to the fact that men generally have a more dismissive attitude towards their own health than women. Men and women have differing attitudes to taking care of their health, to preventive health care, and to health-threatening or anti-health behaviors. Gender stereotypes significantly affect the somatic and mental health of women and men [12].

According to the National Epidemiological Data Center, the average life expectancy in Poland in 2019 was 74.1 years for men and 81.7 years for women, and men are at greater risk for diseases of civilization diseases such as cardiovascular diseases, cancer, and obesity [11]. These diseases are associated with leading an unhealthy lifestyle, alcohol consumption, smoking, consuming a high-fat diet, and low physical activity. Injuries are also more common in men, mainly because of their tendency to engage in risky behaviors [13]. In addition, men are reluctant to attend check-ups, do not follow medical recommendations, and are less likely to seek health care and medical help than women [12].

Epidemiological studies that assess the oral health of adults usually cover specific age ranges [14]. Publications on the dental status of people aged 45–74 are infrequent, and findings are rarely analyzed by sex. There are several publications in the available literature on the dental condition of middle-aged and elderly men; however, groups of male respondents rarely exceed 200 individuals [15,16,17,18,19]. Su et al. [20] evaluated sex differences with regard to oral health and habits in the United States of 4741 subjects (average age was 53.7 years old). Males tended to have fewer dental visits, worse perception of periodontal and dental needs, poorer flossing, and more root caries than women. 

In light of prognoses predicting an increase in human life expectancy and the size of the elderly population (including men), it seems necessary to conduct more research on the issue of men’s oral health. Studies of oral health conditions in aging men—combined with analysis of sociodemographic conditions—may become a fundamental step in the comprehension of factors that determine the oral health of the elderly as well as in the formulation of public health programs. This could support the development of strategies for health-promoting education and sex-specific dental prophylaxis care planning to improve men’s oral health behaviors. 

The aim of this study was to assess the dental status and oral health behaviors in selected 45–74-year-old men from northeastern Poland.

## 2. Materials and Methods

### 2.1. Sample and Setting

The study group consisted of 419 men aged 45–74 years old (mean age 60.0 ± 8.1). They were former employees of an industrial plant—classified as the metal industry—located in the city of Bialystok. This was a group of workers covered by medical care (including dental care) by the industrial health service (1999). After that period, a general practitioner that included a dental office operated at the workplace. Each patient was invited by letter to participate in this study.

### 2.2. Ethical Approval

This study protocol was approved by the local Ethical Committee of the Medical University of Bialystok, Poland (resolution no. R-I-002/245/2010). All participants were informed of the aims and types of this study and provided written consent.

### 2.3. Questionnaire Collection

The data obtained during the survey were entered into a questionnaire, which consisted of the following parts: Personal data;Demographic and socioeconomic factors (age, education level, number of people in household, total monthly household income);Dental history with inclusion of daily oral hygiene habits (frequency of toothbrushing, using additional oral hygiene tools—dental floss, irrigators), and oral health behaviors (frequency of dental appointments, active smoking habit, the number of cigarettes smoked per day, duration of smoking habit);Clinical oral examination;Assessment of oral hygiene.

The digital code made it possible to record the obtained data in Excel and then process statistically.

### 2.4. Oral Examination and Measures

An expert dentist did a clinical evaluation of the patients by performing a thorough examination of the oral cavity to assess oral status. Patients were examined in an office, under artificial lighting using standard instruments (mirror and probe no. 621, recommended by the WHO) [14]. Twenty-eight teeth were considered for clinical examination, omitting third molars [16]. The information obtained during the oral history and clinical examination was used to evaluate:The effectiveness of oral hygiene procedures performed at home by patients using the Approximal Plaque Index (API) according to Lange et al. [21]; the evaluation criterion was the presence (+) or absence (−) of dental plaque in the approximal areas. The API determines the percentage of the sum of the dental plaque surfaces in relation to the sum of all examined areas. The presence of dental plaque was diagnosed in quadrants 1 and 3 of the approximal areas from the oral side and in quadrants 2 and 4 of the approximal areas from the buccal side. The API was determined according to the following scheme: 100–70% = poor, 70–40% = average, 39–25% = rather good, <25% = optimal oral hygiene.The prevalence of caries, meaning the percentage of people affected by dental caries in the study population [14].The intensity of caries in the study population expressed by the mean number of DMFT defined as the sum of Decayed (DT), Missed (MT) and Filled (FT) Permanent Teeth in relation to the number of examined subjects.Mean number of DMFT = (DT + MT + FT)/(number of examined subjects).The following criteria for DMFT were applied [14]:Decayed Tooth (DT)—a tooth with a visible caries cavity or caries lesion detectable by probing; a tooth with a temporary restoration; a tooth with secondary caries;Missing Tooth (MT)—a tooth that has been extracted due to dental caries;Filled tooth (FT)—a tooth with a permanent restoration or covered by a prosthodontic crown and no caries cavity anywhere on the tooth;The percentage of edentulous patients in the study population;The association between demographic and socioeconomic variables and dentition status and oral hygiene; oral and dental hygiene status was correlated with environmental factors: age, education, and per capita income in the family;Oral hygiene and dental status were correlated with health behaviors, i.e., frequency of tooth brushing, frequency of flossing, use of mouthwash, frequency of visiting the dentist, smoking, number of cigarettes smoked per day, and duration of smoking habit.

### 2.5. Data Analysis

Calculations were performed using IBM^®^ SPSS^®^ Statistics software version 20.0 (IBM, Armonk, NY, USA). In descriptive statistics, the mean, median, standard variation, group size and group structure (percentages) were used. 

The relationships between quantitative and ordinal variables were assessed using Spearman’s non-parametric correlations. Non-parametric tests were also used to compare quantitative variables between the subgroups. The Mann–Whitney test was used when comparing two subgroups, and the Kruskal–Wallis test was used when there were more subgroups. All statistical hypotheses were tested at a significance level of α = 0.05.

## 3. Results

### 3.1. Demografic and Socioeconomic Characteristic of the Participants

A total of 419 men, who were invited to participate in this study, answered the questionnaire, underwent clinical examination, and were enrolled. The response rate was 100%. The demographic and socioeconomic characteristics of the participants are summarized in Figure 1, Figure 2 and Figure 3. The male subjects aged 45–74 years (mean age 60.0 ± 8.1 years) were divided into three age groups (Figure 1):–45–54 years (mean age 50.6 ± 2.7),–55–64 years (mean age 58.8 ± 2.6), and–65–74 years (mean age 70.5 ± 2.9).

Among the respondents, the largest percentage was men who had a secondary education (39.9%) and basic vocational education (33.4%) (Figure 2). The average monthly household income per person ranged from PLN 760.40 in the youngest age group to PLN 1012.60 in the oldest (Figure 3).

### 3.2. Dental History

Oral hygiene behaviors are presented in Table 1. A total of 376 respondents underwent assessment of the frequency of tooth brushing and use of additional oral hygiene products was made for 376 respondents as the remaining 43 were edentulous. Most respondents (53.2%) brushed their teeth once a day, while 39.4% brushed twice a day. Only 5.8% of the men brushed their teeth three times a day. Two people brushed their teeth once a week, and four said they did not brush their teeth at all. Only 79 respondents (21%) used additional means to maintain oral hygiene, i.e., mouthwash (10.9%), and dental floss (10.1%). Almost half of the people (45.6%) reported to the dentist for check-ups less than once every two years, while 16.2% of men reported once every two years, and 22.2% once a year. Only 16.0% of respondents went to the dentist once every six months.

The study group’s smoking habit characteristics are shown in Table 2. Active nicotinism affected 26.7% of the male subjects (112 subjects), while 73.3% (307 subjects) reported they were free of the habit. The average number of cigarettes smoked per day in the study group was 16.3. The highest percentage in the study population (57.2%) were men who smoked between 11 and 20 cigarettes per day. The next largest group was those who smoked from 1 to 10 cigarettes per day (32.1%), and more than 21 cigarettes per day were smoked by 10.7% of the total respondents. The average duration of cigarette smoking among all the surveyed was 34.0 years. The largest group (35.7%) was those who had smoked for 31–40 years. The next largest group (27.7%) was those who had smoked cigarettes for 21–30 years. Among the respondents, 21.4% gave a smoking history of more than 40 years, and the shortest period of smoking—up to 20 years—was reported by only 15.2% of the male respondents.

### 3.3. Oral Clinical Examination 

#### 3.3.1. Oral Hygiene

The oral hygiene of the male respondents as assessed by the Approximal Plaque Index (API) is shown in Figure 4. The mean value of the API in the group of 376 men (after excluding 43 edentulous subjects) was 77%. The API values ranged from zero to 100%. The largest group consisted of those with API values ranging 70–100% (247 subjects; 65.7% of men). Sixty-three people (16.8% of the subjects) had sufficient oral hygiene, but with a need for improvement. Fairly good oral hygiene was present in 43 people (11.4% of men), and only 23 (6.1% of subjects) had optimal oral hygiene.

The average API values depending on the men’s age were as follows: 79.3% in the youngest age group (45–54), 76% in the middle age group (55–64) and 75.1% in the oldest (65–74).

#### 3.3.2. Dental Status

Tooth condition expressed by the average number of DMFT and the components DT, MT, FT is shown in Table 3. In the study group, a 100% caries prevalence was observed. The average number of DMFT was 21.4. An average of 4.1 teeth with caries (DT), 4.0 filled (FT), and 13.3 extracted (MT) teeth were found.

The number and percentage of edentulous people by age group in the study population are shown in Table 4. Among the 419 subjects, edentulism was found in 43 people (10.3%). In the youngest age group (45–54 years), there was not a single edentulous person (0%). In the 55–64 age bracket, the percentage of edentulous subjects was 7.9% (14 subjects); while among the oldest subjects, aged 65–74, there were 23.6% of edentulous men (29 subjects).

### 3.4. Associations between Demographic, Socioeconomic Variables and Health Behavior, Oral Hygiene, and Dentition Status 

#### 3.4.1. Oral Hygiene

The associations between demographic, economic and health behavioral characteristics and oral hygiene status are shown in Table 5. As the subjects’ per capita family income increased, the average API value decreased significantly (r = −0.12; *p* = 0.024)—a negative correlation. Among the male respondents who brushed their teeth once a day, there was a significantly higher mean API value compared with those brushing two (*p* = 0.012) or three times a day (*p* < 0.001). At the same time, among the respondents who brushed their teeth twice a day, a significantly higher mean API value was also observed than among the men who brushed three times a day (*p* = 0.019).

Among the men who flossed their teeth, a significantly lower average API value was found compared with subjects who did not floss (*p* < 0.001). Among male smokers, a significantly higher mean API value was found than in the nonsmoking group (*p* = 0.027). As the number of cigarettes smoked per day and the duration of the smoking habit increased, the mean API value increased significantly, indicating a positive correlation between these variables (*p* = 0.027 and *p* = 0.026, respectively).

There were no statistically significant correlations between the mean API values and the subjects’ age (*p* = 0.091) and education (*p* = 0.317), as well as their use of mouthwash (*p* = 0.581) and frequency of visits to the dentist (*p* = 0.585).

#### 3.4.2. Dental Status

The correlations between demographic, economic and health behavioral characteristics and dental status are shown in Table 6. As the men’s ages increased, the mean MT value (*p* < 0.001) and mean DMFT (*p* < 0.001) significantly increased, indicating a positive correlation between these variables, while mean DT (*p* < 0.001) and FT (*p* < 0.001) numbers significantly decreased—a negative correlation.

As the educational level of the subjects increased, the mean FT number significantly rose (*p* < 0.001), indicating a positive correlation between these variables, while mean MT and DMFT significantly decreased (*p* < 0.001)—a negative correlation. There was no significant correlation between the level of education and mean DT number (*p* = 0.051), but the obtained result remained on the borderline of significance.

As the subjects’ per capita family income increased, the average DMFT number rose significantly (*p* = 0.03), indicating a positive correlation. There was no significant correlation between the per capita family income of the subjects and the mean DT, MT, and FT (*p* > 0.05).

As the frequency of tooth brushing increased, the average number of filled teeth—FT rose significantly (*p* < 0.001), indicating a positive correlation between these variables. There was no significant correlation between the frequency of tooth brushing and mean DT, MT, and DMFT values (*p* > 0.05).

Among those who flossed, significantly higher mean DT (*p* = 0.02) and FT (*p* < 0.001), and significantly lower mean MT (*p* < 0.001) and DMFT (*p* = 0.01) were observed compared with other men who did not floss. 

As the API increased, the mean DT (*p* = 0.02), MT (*p* < 0.001), and DMFT (*p* = 0.049) significantly rose, indicating a positive correlation between these variables, while mean FT (*p* < 0.001) significantly decreased (negative correlation).

With an increase in the frequency of the respondents’ visits to the dentist, the average numbers of MT (*p* < 0.001) and DMFT (*p* < 0.001) were significantly reduced (negative correlation). At the same time, as the frequency of follow-up visits increased, the mean number of FT increased significantly (*p* < 0.001)—a positive correlation. There was no significant correlation between the frequency of reporting to the dentist and the average number of DT (*p* = 0.06).

Those who smoked cigarettes had a significantly lower mean FT number compared with the other subjects (*p* < 0.001). In those who smoked cigarettes, the average FT number was significantly lower compared with the other subjects (*p* < 0.001). There was no significant correlation between cigarette smoking and the mean DT, MT, and DMFT (*p* > 0.05). As the number of cigarettes smoked per day increased, the average FT number decreased significantly (*p* < 0.001)—a negative correlation. There was no significant correlation between the number of cigarettes smoked per day and the mean DT, MT and DMFT numbers (*p* > 0.05). With the lengthening of the period of cigarette smoking among the studied men, the average number of MT increased significantly (*p* = 0.03)—a positive correlation, and the average number of FT decreased significantly (*p* < 0.001)—a negative correlation. There was no significant correlation between the duration of cigarette smoking and mean DT and DMFT (*p* > 0.05). The mean DT, MT, FT and DMFT numbers in men using mouthwash and others were not statistically significantly different (*p* > 0.05).

## 4. Discussion

### 4.1. Oral Hygiene

The results of our study indicate that the studied patients’ inadequate state of oral hygiene—the average API of the toothed men was 77%. In other studies conducted in Poland among men aged 65–74, the average API ranged from 67 to 75.8%, which is lower or similar to the obtained results [22,23]. Particularly noteworthy is the highest percentage of men (65.7%) characterized by improper oral hygiene, despite the fact that they were notified by letter about the scheduled dental examination. Similar results were obtained among senior citizens (aged 60–75) from Salvador using the simplified Oral Hygiene Index (OHI), where the percentage of men with poor oral hygiene was 67.8% [19].

Butera et al. [8] showed that the use of basic and supplementary measures to maintain oral hygiene along with frequent recalls of oral hygiene had a positive impact on improving the oral health status of patients with diabetes and their life style.

In the present study, we found that more frequent tooth brushing and flossing were significantly associated with better oral hygiene. A similar trend was also confirmed in previous studies [24,25]. On the other hand, among Lithuanian senior residents, the effect of oral hygiene on the rate of tooth loss was not confirmed. This was most likely due to the fact that the respondents lived in an area with high fluoride content in the environment. In the same study, the association of oral hygiene with the average number of fillings was consistent with our observations [26].

Ageing comes with difficulties in maintaining oral hygiene and the following bias factors may be included reduced manual skills, general disability of the elderly, visual defects, removable prostheses in the mouth, decreased saliva production, and missing teeth [9].

### 4.2. Dental Status

Our own study found a 100% prevalence of dental caries in the entire study group of men as well as in three separate age subgroups. A study by Barczak [27] conducted in the West Pomeranian region also found a 100% caries prevalence in a group of men and women over the age of 55. The results are similar to those obtained by other researchers who evaluated caries prevalence among people over 45 years of age [17,18,28].

In the presented results, the mean DMFT increased with age. The highest number among the DMFT components was the average MT, which significantly influenced the high DMFT index. The other components—the average decayed and filled teeth—decreased as the age of the men increased, although this decrease was not as pronounced as the average MT. Epidemiological studies on the oral health of men over 45 years of age over recent years have also obtained the most significant component as the average MT number but DMFT ranged from 11.9 ± 8.59 to 26.9 ± 6.3 [15,16,17,18,19,29]. In addition, our study showed that there was a significant positive correlation between age and the average DMFT and MT numbers, while there was a negative correlation between DT and FT numbers. This confirms the existence of a positive correlation between age and the intensity of caries and tooth loss [18,19,29,30,31,32,33]. 

### 4.3. Edentulism

Among the examined respondents, the prevalence of edentulism increased with age, which is consistent with the observations of other authors [34]. In the youngest age group of 45–54 years, no edentulousness was found; but in the oldest subgroup aged 65–74 years was near 25%. In other countries similar data were observed among Danes (9%) of both sexes [33] and among men from north-eastern Germany (11.1%) [35]. Lower percentages of edentulousness were noted by Nakayama et al. [16] among men over 50 (1.4%), Vitosyte et al. [29] in a study of men and women aged 35–74 (3.8%), Ngujena et al. [17] in patients of both sexes aged 65–74 (5.8%), and Valasquez-Olmedo et al. [36] in older adults with a mean age of 68.1 years (6.9%). Higher percentages of edentulous individuals than those obtained in our own work were reported by Stojanovic et al. [15] in a group of 88 men aged 65–74 (27.3%), Aquirre Escobar et al. [19] in Salvadorans aged 60 years and older (almost one-third of the participants), and Carneiro de Oliveira et al. [37] in male and female individuals over 60 years (50.8%). 

The following determinants may play a role: geographic region, economic development of the country, age, sex, education, income, comorbidity, and lifestyle [30,38]. Toothlessness tends to be higher among women than men, but this difference is gradually being blurred [30,38]. The reason for the higher frequency of extractions among women is the disruption of calcium metabolism during pregnancy and menopause [39]. Additionally, there is a higher caries prevalence among women than men, which is attributed to earlier eruption of teeth in girls potentially extending exposure to cariogenic oral factors, hormonal influences, and genetic predisposition, such as variants of the X-linked amelogenin gene [40], as well as the nature of group selection (random or purposive).

The prevalence of edentulism, both in Poland and in the northeast region, is gradually decreasing, but still remains one of the highest in Europe [41]. This is most likely due the National Health Fund’s failure to finance endodontic treatment of molars and premolars and the lack of refunds for the reconstruction of missing teeth with fixed prosthetic dentures. Removable partial dentures are an iatrogenic etiological factor of periodontitis, and their use consequently leads to the extraction of teeth that limit the space between missing teeth in the dental arch. Patients are more likely to opt for surgical treatment, and the dental care system underestimates the effectiveness of preventive care.

### 4.4. Associations between Demographic, Socioeconomic Variables and Health Behavior, Oral Hygiene, and Dentition Status

#### 4.4.1. Socioeconomic Factors vs. Dental and Oral Hygiene Status

Among the socioeconomic factors that are determinants of health, the education and material status of individual members of the community take the lead. Particularly relevant is the level of education, which is the most important single variable affecting health attitudes and behavior [42]. This is supported by our study, with regard to the impact of socioeconomic status on dental caries and oral hygiene status. In the study group of men, we found that as the level of education increased, the average number of DMFT and the MT component decreased significantly. In addition, the FT component increased significantly with a rising education level.

Many authors confirmed our observations regarding the negative correlation between education level and the average DMFT number [18,43,44] and its MT component [22,31,32,33,45] and the positive correlation between education and the FT component [22,23,27]. 

Numerous studies suggest that high material status tends to lower caries intensity and teeth extractions, since restorative treatment usually requires financial resources [22,31,32,33,37,45,46]. 

An interesting observation was made among Norwegian adults. Researchers found that socioeconomic differences, which usually appear at an early age and affect the value of the DMFT number, cause it to remain unchanged for the next 30 years, even if the differences in the group decrease during this period [47]. Socioeconomic status is an important indicator an increased risk of caries intensity in societies, so caries prevention programs should be differentiated and adapted to the level of economic development of a country [43,48]. 

In the analyzed group, there was no statistically significant correlation between the mean API and the level of education, while per capita family income differed significantly. As the per capita family income of the subjects increased, the men’s oral hygiene significantly improved. However, some authors have noted a significant beneficial effect of both characteristics—education and higher income—on oral hygiene [22,31]. Positive health behaviors, such as the use of additional oral care products, regular toothbrushing and dental check-up visits, are more frequently observed among respondents with a higher education and income [18,22,44]. In our study, all participants (both those with higher and lower per capita family income) were provided with free dental care through the industrial health service, which may explain some of the differences.

#### 4.4.2. Smoking Habits and Oral Conditions

In the current study, slightly higher mean DMFT numbers and their components, DT and MT, were observed among male smokers, but the obtained differences were not statistically significant, while significantly lower FT numbers were registered. In addition, there was no significant correlation between the mean DMFT and DT numbers and active smoking, the number of years of smoking and the number of cigarettes smoked per day. However, a significant positive correlation was found between the mean MT and smoking duration. This suggests that the longer the subjects smoked, the fewer teeth they had. In previous studies habitual smoking was significantly associated with the risk of having fewer teeth in the male and female subjects [16,31,32,45,49].

In addition, our study showed that male smokers were characterized by a significantly higher mean API than nonsmokers. A positive correlation between the average API, duration of smoking, and the number of cigarettes smoked per day was observed. Other authors also showed significant covariances in relation to the discussed correlations [22,50]. However, other researchers claim that the intensity of any negative changes in the oral cavity associated with smoking does not depend on the number of cigarettes smoked, but on the duration of smoking and on oral hygiene level [51].

The effect of smoking on the oral cavity is undoubtedly negative and causes several chemical, physical, and mechanical side effects. Moreover, smoking causes a decrease in salivary secretion (sialopenia) and its buffering capacity, a dry mouth and promoting greater susceptibility to caries [51,52]. In addition, nicotine has been proven to enhance the adherence of Streptococcus mutans [53] and promotes the cariogenic activity of oral microorganisms with the formation of a caries-susceptible environment [54]. Vitosyte et al. [29] assessed the determinants associated with oral health conditions among adult residents aged 35–74 years old in Vilnius, Lithuania and showed that smoking frequency was significantly associated with dental caries. However, Ijang et al. [53] in a recent systematic review and meta-analysis claimed that, there was insufficient evidence to confirm the hypothesis that tobacco, as a risk factor, is involved in the dental caries process.

### 4.5. The World Health Organization’s Oral Health Goals and Strategies for the Elderly Population

The WHO’s oral health goals for the year 2020 in the elderly were formulated as [55]:Reduce the number of teeth extracted because of dental caries at age 65–74 years by X%; Reduce the number of teeth lost because of periodontal diseases by X% at age 65–74 years with special reference to tobacco use, poor oral hygiene, stress and inter-current systemic diseases;Reduce the number of edentulous persons by X% at age 65–74 years, increase the number of natural teeth present by X%, and increase the number of individuals with functional dentitions (20 or more natural teeth) by X% at age 65–74 years. 

Comparing the results of our study with those obtained in the health monitoring of the Polish population carried out in 2019 in men aged 65–74, the planned 2020 oral health goals to reduce the number of teeth extracted due to caries were achieved within 10 years from MT = 21.6 (2009) to MT = 17.6 (present study). However, the average number of MT in our study was slightly higher than in the 2019 monitoring study (MT = 16.0) [5]. In addition, in our study, the percentage of edentulous men was also reduced to 21.43% from 39.6% (2009). On the other hand, the DMFT value decreased slightly during the observation period, remaining at a very high level (2009 = 24.8; the present study = 21.43; monitoring 2019 = 25.0) [5].

Despite a noticeable improvement over the past decade, the oral health of Polish seniors aged 65–74 is still unsatisfactory. Although the availability of dental treatment for the elderly in Poland is relatively high, the actual attainability of medical services is limited, as evidenced by the high percentages of people with edentulism (above 20%) and periodontal pockets requiring specialized treatment (44.8%) and the low number of present natural teeth (12.5) (monitoring 2019) [5]. 

The ageing of the population and an increase in the percentage of seniors in the Polish population is associated with a rise in health problems. This situation emphasizes the need to intensify preventive and educational activities aimed at the elderly and their caregivers. Implementation of this project requires the involvement of the entire medical community—family physicians, geriatricians, and dental teams—with particular emphasis on the role of dental hygienists.

In Poland, most dental hygienists are employed to provide basic preventive and therapeutic procedures such as oral hygiene instruction, fluoridation, dental scaling and polishing, although they have many more competencies and skills. To provide more effective dental care among seniors in Poland, dental hygienists should be more involved in conducting oral health education and promotion among both seniors and non-dental health care professionals (caregivers, nurses, general practitioners) noting that the most common chronic diseases (heart disease, stroke, cancer) have identical risk factors (poor nutrition, inadequate oral hygiene, smoking, and stress) with dental caries and periodontal disease.

A change in the population-wide approach to disease prevention seems necessary. The dental care system should focus more on promoting oral health and achieving greater equality in access to health services. A noteworthy proposal is the establishment of dental facilities dedicated to senior patients, which would operate under preferential state contracting for dental services.

Sugar, alcohol and tobacco consumption and underlying social and commercial factors contribute to the development of diseases, including oral diseases. Comprehensive regulations seem necessary to reduce the negative consequences of adverse civilization-related changes affecting society’s health. 

The WHO Global Oral Health Program has defined the main strategies for improving older people’s oral health of older people through oral health policy, oral health care, education and training for service and care, and research [56]. Moreover, the WHO guidelines on Integrated Care for Older People (ICOPE) indicates evidence-based recommendations for health care professionals to prevent, slow or reverse declines in the physical and mental capacities of older people. These recommendations require countries to place the needs and preferences of older adults at the center and to coordinate care. The ICOPE guidelines will allow countries to improve the health and well-being of their older populations, and to move closer to the achievement of universal health coverage for all at all ages [57].

### 4.6. Study Limitations 

A limitation of this study is that the selection of the study population was not random but intentional. The study population consisted of men who were employees of an industrial plant in Bialystok. Therefore, the obtained results may not reflect the actual oral health status and health behavior of men living in the northeastern region of Poland. A further limitation is the failure to take into account in the current paper the relationship between systemic diseases and the dental status of the subjects, as well as the status of the oral mucosa and possible malignant neoplastic lesions that would be expected in a population of smoking men. These essential issues should be addressed in the futured studies. Moreover, there is an urgent need for systematic, standardized, and reproducible epidemiological studies in the older adults.

## 5. Conclusions

The topic of the dental status and oral hygiene habits of men aged 45–74 is seldom covered in the epidemiological studies on the oral health condition; therefore, our results may update missing data. Unfortunately, the current study demonstrated a low health awareness among the surveyed men manifested by inadequate oral hygiene habits, a high average API and infrequent visits to dentist. Despite good access to dental services, the subjects’ oral health was unsatisfactory, as evidenced by the high mean DMFT and a significant percentage of edentulous subjects, especially in the oldest age group. In the studied male population, correlations were found between demographic, socioeconomic and health behavioral characteristics and dental and oral health status. The poor oral health condition of the study population indicates the need to intensify pro-health education among seniors about oral care. Evaluating the oral health condition of older adults connected with sociodemographic factors and oral health behaviors is essential for developing public policies and effective oral health strategies addressed to this population. 

## Figures and Tables

**Figure 1 ijerph-20-06005-f001:**
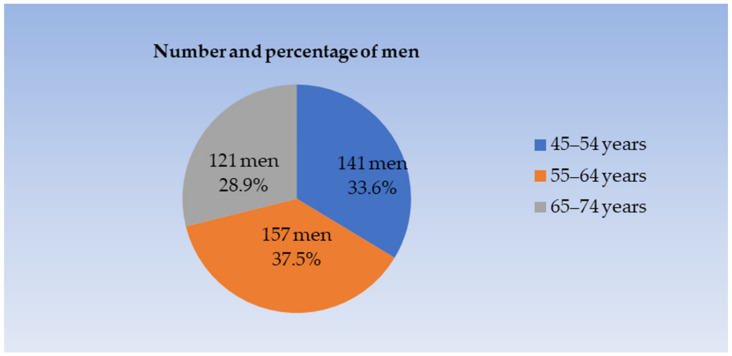
The distribution of respondents based on the age characteristics of study participants.

**Figure 2 ijerph-20-06005-f002:**
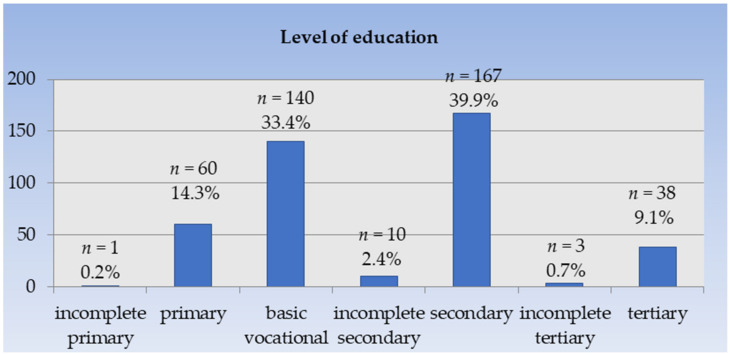
Distribution of subjects based on the education level. *n*—number of male participants.

**Figure 3 ijerph-20-06005-f003:**
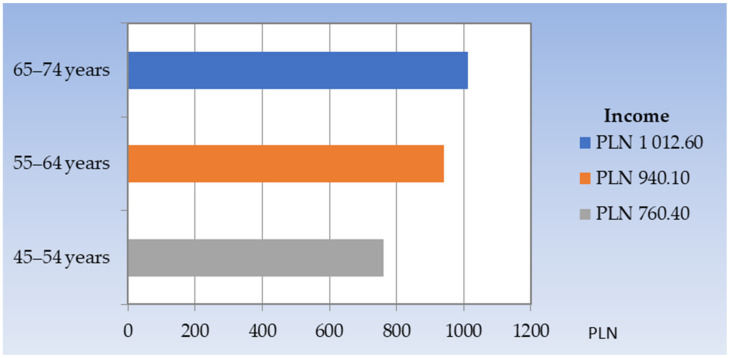
Distribution of the average monthly income per person in the household by the age group.

**Figure 4 ijerph-20-06005-f004:**
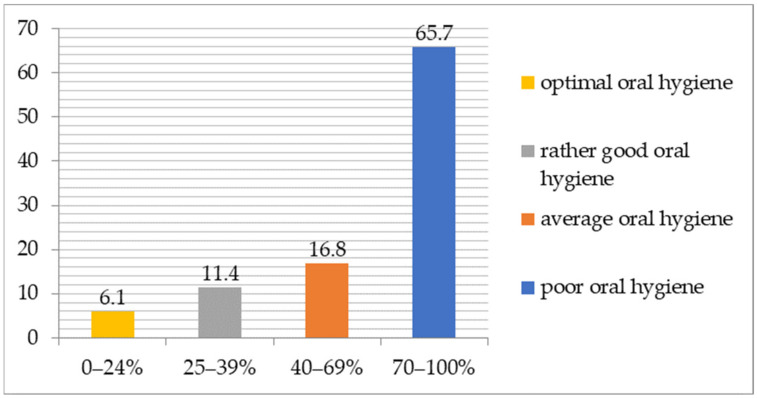
The API in the study population (*n* = 376).

**Table 1 ijerph-20-06005-t001:** Number and percentage of subjects depending on the frequency of tooth brushing (*n* = 376), using additional oral hygiene methods (*n* = 376) and dental appointments (*n* = 419).

Dental Interview			
		*n*	%
Frequency of tooth brushing	not at all	3	0.8
	once a week	2	0.5
	once a day	201	53.5
	twice a day	148	39.4
	three times a day	22	5.8
	total (dentulous)	376	100.0
	edentulous	43	-
Additional methods of oral hygiene	use of dental floss	38	10.1
	use of mouthwashes	41	10.9
	total (dentulous)	376	100.0
	edentulous	43	-
Frequency of dental appointments	less than once every two years	191	45.6
	once every two years	68	16.2
	once a year	93	22.2
	once every half year	67	16.0
	total	419	100.0

*n*—number of people; %—percentage.

**Table 2 ijerph-20-06005-t002:** Characteristics of nicotinism in the study population (*n* = 419).

Variables	Characteristics of Nicotinism					
		*n*	%	*n**	$\bar{X}$	SD
Smoking	yes	112	26.7			
	no	307	73.3	–	–	–
	total	419	100.0			
Number of cigarettes smoked per day	1–10 pcs	36	32.1	112	16.3	8.6
	11–20 pcs	64	57.2			
	>21 pcs	12	10.7			
	total	112	100.00			
Duration of smoking habit in years	1–20	17	15.2	112	34.0	11.2
	21–30	31	27.7			
	31–40	40	35.7			
	>40	24	21.4			
	total	112	100.0			

*n*—number of people, *n**—number of smoking people, %—percentage, and $\bar{X}$—mean, SD—standard deviation.

**Table 3 ijerph-20-06005-t003:** Caries status of participants expressed by DMFT and its components.

	DT	MT	FT	DMFT
*n*	419	419	419	419
$\bar{X}$	4.1	13.3	4.0	21.4
SD	3.8	8.4	3.9	5.5
Me	3.0	12.0	3.0	22.0

*n*—number of people, $\bar{X}$—mean, SD—standard deviation, and Me—median.

**Table 4 ijerph-20-06005-t004:** Percentage of edentulous patients in the study groups depending on the age.

Age	Size Group	Edentulous	
		*n*	%
45–54 years	118	0	0
55–64 years	178	14	7.9
65–74 years	123	29	23.6
Total	419	43	10.3

*n*—number of people.

**Table 5 ijerph-20-06005-t005:** Mean API depending on sociodemographic factors and pro- and anti-health behaviors.

	Variables/Items	API
		$\bar{X}$ ± SD
Socio demographic factors	age groups	
	45–54 (*n* = 118)	79 ± 30
	55–64 (*n* = 164)	76 ± 30
	65–74 (*n* = 94)	75 ± 30
	*p****	0.091
	education level	
	incomplete primary (*n* = 1)	100
	primary (*n* = 49)	80 ± 30
	basic vocational (*n* = 126)	78 ± 30
	incomplete secondary (*n* = 10)	64 ± 40
	secondary (*n* = 151)	77 ± 30
	incomplete tertiary (*n* = 3)	85 ± 20
	tertiary (*n* = 36)	68 ± 30
	*p****	0.317
	per capita family income	
	r*	−0.12
	*p**	0.024
Pro-health behaviors	frequency of tooth brushing	
	not at all (*n* = 3)	94 ± 100
	once a week (*n* = 2)	100 ± 0.0
	once a day (*n* = 201)	82 ± 30
	twice a day (*n* = 148)	73 ± 30
	three times a day (*n* = 22)	50 ± 40
	*p****	<0.001
	frequency of dental visits	
	less than once every two years (*n* = 149)	78 ± 30
	once every two years (*n* = 68)	80 ± 30
	once a year (*n* = 93)	75 ± 30
	once every half year (*n* = 66)	74 ± 30
	*p****	0.585
	use of mouthwashes	
	yes (*n* = 41)	74 ± 30
	no (*n* = 335)	77 ± 30
	*p***	0.581
	use of dental floss	
	yes (*n* = 38)	62 ± 30
	no (*n* = 338)	78 ± 30
	*p***	<0.001
Anti-health behaviors	smoking	
	yes (*n* = 97)	82 ± 30
	no (*n* = 279)	75 ± 30
	*p***	0.027
	number of cigarettes smoked per day	
	r*	0.11
	*p**	0.027
	duration of smoking habit in year	
	r*	0.12
	*p**	0.026

*n*—number of men, $\bar{X}$—mean, SD—standard deviation, r*—Spearman’s rank correlation coefficient, *p**—Spearman’s rank correlation coefficient test, and *p***—Mann–Whitney test. *p****—Kruskal–Wallis test.

**Table 6 ijerph-20-06005-t006:** Dentition status expressed by the mean number of DMFT depending on sociodemographic factors, pro- and anti-health behaviors.

Variables/Items		DMFT	DT	MT	FT
		$\bar{X}$ ± SD	$\bar{X}$ ± SD	$\bar{X}$ ± SD	$\bar{X}$ ± SD
Socio demographic factors	age groups				
	45–54 (*n* = 118)	18.1 ± 4.8	5.4 ± 3.5	7.7 ± 5.6	5.0 ± 3.5
	55–64 (*n* = 178)	21.9 ± 5.1	4.2 ± 3.8	13.9 ± 7.9	3.8 ± 3.6
	65–74 (*n* = 123)	23.8 ± 5.2	2.8 ± 3.6	17.6 ± 8.3	3.4 ± 4.3
	r*	0.41	−0.32	0.45	−0.22
	*p**	0.001	<0.001	<0.001	<0.001
	education level				
	incomplete primary (*n* = 1)	20.0	5.0	7.0	8.0
	primary (*n* = 60)	23.8 ± 4.4	3.5 ± 3.3	17.4 ± 8.39	2.9 ± 3.7
	basic vocational (*n* = 140)	21.7 ± 5.5	3.7 ± 3.4	14.3 ± 8.1	3.7 ± 3.5
	incomplete secondary (*n* = 10)	21.5 ± 5.4	5.9 ± 5.5	12.7 ± 6.5	2.9 ± 3.1
	secondary (*n* = 167)	20.5 ± 5.6	4.6 ± 4.2	11.4 ± 8.1	4.5 ± 3.9
	incomplete tertiary (*n* = 3)	20.3 ± 8.3	2.3 ± 4.0	14.3 ± 9.8	3.7 ± 3.7
	tertiary (*n* = 38)	20.6 ± 5.1	4.4 ± 3.3	10.0 ± 7.7	6.2 ± 4.1
	r*	−0.19	0.10	−0.26	0.20
	*p**	<0.001	0.051	<0.001	<0.001
	per capita family income				
	r*	0.11	−0.04	0.09	0.01
	*p**	0.03	0.42	0.08	0.87
Pro-health behaviors	frequency of tooth brushing				
	not at all (*n* = 3)	18.3 ± 5.5	5.7 ± 1.1	11.3 ± 3.7	1.3 ± 2.3
	once a week (*n* = 2)	22.5 ± 3.54	2.5 ± 3.5	18.5 ± 4.9	1.5 ± 2.1
	once a day (*n* = 201)	20.8 ± 5.5	4.7 ± 3.9	12.1 ± 7.3	4.0 ± 3.8
	twice a day (*n* = 148)	20.5 ± 5.11	4.4 ± 3.25	10.8 ± 6.87	5.3 ± 3.7
	three times a day (*n* = 22)	20.5 ± 5.7	4.7 ± 5.0	11.4 ± 7.7	4.4 ± 3.7
	r*	−0.04	−0.03	−0.08	0.17
	*p**	0.46	0.58	0.11	<0.001
	API				
	r*	0.10	0.12	0.16	−0.28
	*p**	0.049	0.02	<0.001	<0.001
	frequency of dental visists				
	less than once every two years (*n* = 191)	22.2 ± 5.9	3.9 ± 4.3	16.1 ± 9.1	2.2 ± 3.1
	once every two years (*n* = 68)	20.6 ± 5.7	4.5 ± 3.3	11.5 ± 7.4	4.6 ± 3.6
	once a year (*n* = 93)	20.6 ± 4.5	4.4 ± 3.3	10.7 ± 6.9	5.5 ± 3.5
	once every half year (*n* = 67)	21.0 ± 5.1	3.8 ± 3.1	10.5 ± 6.2	6.7 ± 3.9
	r*	−0.14	0.09	−0.27	−0.50
	*p**	<0.001	0.06	<0.001	<0.001
	use of mouthwashes				
	yes (*n* = 41)	21.2 ± 4.8	4.8 ± 4.3	11.4 ± 7,2	5.0 ± 4.1
	no (*n* = 335)	21.4 ± 5.6	4.0 ± 3.7	13.5 ± 8.5	3.9 ± 3.8
	*p***	0.64	0.28	0.21	0.09
	use of dental floss				
	yes (*n* = 38)	19.3 ± 4.5	5.0 ± 3.2	6.6 ± 4.7	7.7 ± 3.9
	no (*n* = 338)	21.6 ± 5.5	4.0 ± 3.8	13.9 ± 8.4	3.7 ± 3.6
	*p***	0.01	0.02	<0.001	<0.001
Anti-health behaviors	smoking				
	yes (*n* = 112)	21.7 ± 5.9	4.7 ± 4.2	14.5 ± 8.7	2.5 ± 3.0
	no (*n* = 307)	21.3 ± 5.3	3.9 ± 3.6	12.8 ± 8.2	4.6 ± 4.0
	*p***	0.33	0.13	0.11	<0.001
	number of cigarettes smoked per day				
	r*	0.04	0.09	0.07	−0.24
	*p**	0.37	0.08	0.14	<0.001
	duration of smoking habit in years				
	r*	0.07	0.05	0,11	−0.24
	*p**	0.14	0.27	0.03	<0.001

*n*—number of people, $\bar{X}$—mean, SD—standard deviation, r*—Spearman’s rank correlation coefficient, *p**—Spearman’s rank correlation coefficient test, and *p***—Mann–Whitney test.

## Data Availability

The data presented in this study are available on request from the second author.

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
