# Peer review of "Dental Status and Oral Health Behaviors of Selected 45–74-Year-Old Men from Northeastern Poland"

_ijerph, 2023, doi:10.3390/ijerph20116005_

Round 1
Reviewer 1 Report
The title of the paper Poor oral health status and behaviors in 45-74-year-old men from north-eastern Poland is not appropriate because it does not make clear the purpose of the work.
In the abstract, the conclusion is not clearly defined.
In the introductory part, the focus is mainly on the aging of the population, while little is said about the basic problems of oral health in the elderly population.
It is not clear whether the 100% response rate reported in the results refers to respondents who also underwent a clinical examination after completing the questionnaire or whether it refers to the response of participants who were invited to participate in the study by letter.
The English translation is quite poor, so some sentences lose their meaning or are difficult to understand. Thera are many grammatical errors, too.
Figure 5 is redundant because there is no statistically significant difference between age groups.
The discussion is well written, but the conclusion is inadequate. The discussion should refer to the World Health Organization's proposed oral health goals and strategies for the elderly population. The concluding sentences are not clear; they need to be reworded to make sense of the entire paper and to relate the results obtained to possible clinical or public health goals.
The quality of the English language is not satisfactory. Some sentences are difficult to understand, and many of them lose their meaning because of the poor translation.
Reviewer 2 Report
I think the authors should add to the introduction a short paragraph detailing previous studies conducted in this line of research. The authors reviewed them in the discussion. However, I believe they should have been mentioned earlier.
Furthermore, I advise including managerial implications for authorities (e.g., oral health education, free toothbrush kits, free dental exam once a year), detailing the study limitations, and providing directions for future research.
Readers can easily understand the study’s aims, results, and conclusions. However, I recommend improving the writing to make the paper more eloquent and the style sharper.
Reviewer 3 Report
The manuscript is well written in general, introduction gives a coherent background, methods are well explained and data analysis was correctly performed. However, the subject of study is a population with very similar socioeconomic profile which comprises the obtained results. Different populations in Poland should be included to really draw a profile of north eastern men in Poland and their oral habits. In addition, in a population with numerous smoker individuals and poor hygiene, as the studied group, the presence of potentially malignant lesions of oral mucosa, should be assessed. Finally, text title suggests that the influence of oral health status in subjects' behaviour was the object of study.
Reviewer 4 Report
Improve the introduction explaining why your study is so important.
Remember the importance of the correlation between oral status and diabetes.
Butera A, Lovati E, Rizzotto S, Segu' M, Scribante A, et Al. Professional and home-management in non-surgical periodontal therapy to evaluate the percentage of glycated hemoglobin in type 1 diabetes patients. International Journal of Clinical Dentistry 2021:(14)1:41-53.
Line 73 plants what do you mean?
Methods Did you asked about general health?
Line 126 per capita? pro capite
Discuss in the conclusions the role of the dental hygienist and poor role of the insurance.
Is there the dental hygienist in Poland?
English must improved
Round 2
Reviewer 1 Report
The overall quality of the paper has been significantly improved so I find it appropriate for publishing.
Reviewer 3 Report
Manuscript has been widely revised, title and text are now connected, and limitations of the study are well described.
English language was revised and text reading was improved as well, minor editing is necessary.